# Association of epicardial fat with cardiac structure and function and cardiovascular outcomes: A protocol for systematic review and meta-analysis

Hidekatsu Fukuta[1]*, Toshihiko Goto[2], Takeshi Kamiya[3]

1 Core Laboratory, Nagoya City University Graduate School of Medical Sciences, Nagoya, Japan,
2 Department of Cardiology, Nagoya City University Graduate School of Medical Sciences, Nagoya, Japan,
3 Department of Medical Innovation, Nagoya City University Graduate School of Medical Sciences, Nagoya, Japan

* fukuta-h@med.nagoya-cu.ac.jp

## Abstract

### Background

Epicardial fat represents visceral adiposity. Many observational studies have reported that increased epicardial is associated with adverse metabolic profile, cardiovascular risk factors, and coronary atherosclerosis in patients with cardiovascular diseases and in general population. We and others have previously reported the association of increased epicardial fat with left ventricular (LV) hypertrophy and diastolic dysfunction as well as the development of heart failure (HF) and coronary artery disease in these populations. In some studies, however, the association did not reach statistical significance. The inconsistent results may be due to limited power, different imaging modalities for quantifying epicardial fat volume, and different outcome definitions. Accordingly, we aim to perform the systematic review and meta-analysis of studies on the association of epicardial fat with cardiac structure and function and cardiovascular outcomes.

### Methods

This systematic review and meta-analysis will include observational studies examining the association of epicardial fat with cardiac structure and function or the cardiovascular outcomes. Relevant studies will be identified by searching electronic databases including PubMed, Web of Science, and Scopus and by manual screening of reference lists of relevant reviews and retrieved studies. The primary outcome will be cardiac structure and function. The secondary outcome will be cardiovascular events including death from cardiovascular causes, hospitalization for HF, nonfatal myocardial infarction, and unstable angina.

### Discussion

The results of our systematic review and meta-analysis will provide evidence regarding the clinical usefulness of epicardial fat assessment.

**Data Availability Statement:** No datasets were generated or analysed during the current study. All relevant data from this study will be made available upon study completion.

**Funding:** The funders had and will not have a role in study design, data collection and analysis, decision to publish, or preparation of the manuscript.

**Competing interests:** The authors have declared that no competing interests exist.

**Abbreviations:** HF, heart failure; LV, left ventricular; NP, natriuretic peptide; PRISMA-P, Preferred Reporting Items for Systematic Review and Meta-analysis Protocols.

## Systematic review registration

INPLASY 202280109.

## Introduction

Epicardial fat represents visceral adiposity. Many observational studies have reported that increased epicardial is associated with adverse metabolic profile, cardiovascular risk factors, and coronary atherosclerosis in patients with cardiovascular diseases and in general population [1]. We and others have previously reported the independent association of increased epicardial fat with left ventricular (LV) hypertrophy and diastolic dysfunction as well as the development of heart failure (HF) and coronary artery disease in these populations [2–8]. In some studies, however, the association did not reach statistical significance [9, 10]. The inconsistent results may be due to limited power, different imaging modalities for quantifying epicardial fat volume (echocardiography, CT, or MRI), and different outcome definitions. Accordingly, we aim to perform the systematic review and meta-analysis of studies on the association of epicardial fat with cardiac structure and function and cardiovascular outcomes.

## Methods

This study has been registered on International Platform of Registered Systematic Review and Meta-analysis Protocols with registration number of INPLASY 202280109 (https://www.doi.org; DOI: 10.37766/inplasy2022.8.0109). This protocol for meta-analysis will be performed according to the Preferred Reporting Items for Systematic Review and Meta-analysis Protocols (PRISMA-P) statement [11].

### Search strategy

The electronic databases for literature search will include PubMed, Web of Science, and Scopus. For search of the eligible studies, the following keywords and Medical Subject Heading will be used:

#1 "epicardial adipose tissue" OR "epicardial fat" OR "pericardial adipose tissue" OR "pericardial fat" OR "cardiac adipose tissue" OR "cardiac fat" OR "subepicardial adipose tissue" OR "subepicardial fat" OR "heart fat" OR "heart adipose tissue"

#2 "left ventricular" OR "left ventricle" OR "systolic function" OR "systolic dysfunction" OR "diastolic function" OR "diastolic dysfunction" OR "left atrium" OR "left atrial"

#3 "cardiovascular" OR "hospitalization" OR "heart failure" OR "myocardial infarction" OR "unstable angina" OR "prognosis"

#4 #1 AND #2 (primary outcome)

#5 #1 AND #3 (secondary outcome)

Literature search will also be conducted by manual screening of reference lists of relevant reviews and retrieved articles. Two researchers (HF and TG) will independently perform the literature search. Disagreements will be resolved by consensus. The literature search will be repeated before completing the data extraction, and potential new studies published during the work process will be added to the result. Study selection will be conducted in a PRISMA-compliant flow chart (Fig 1). Only articles published in the English language will be included.

### Study design

Observational studies will be included. Case-control studies will be excluded.

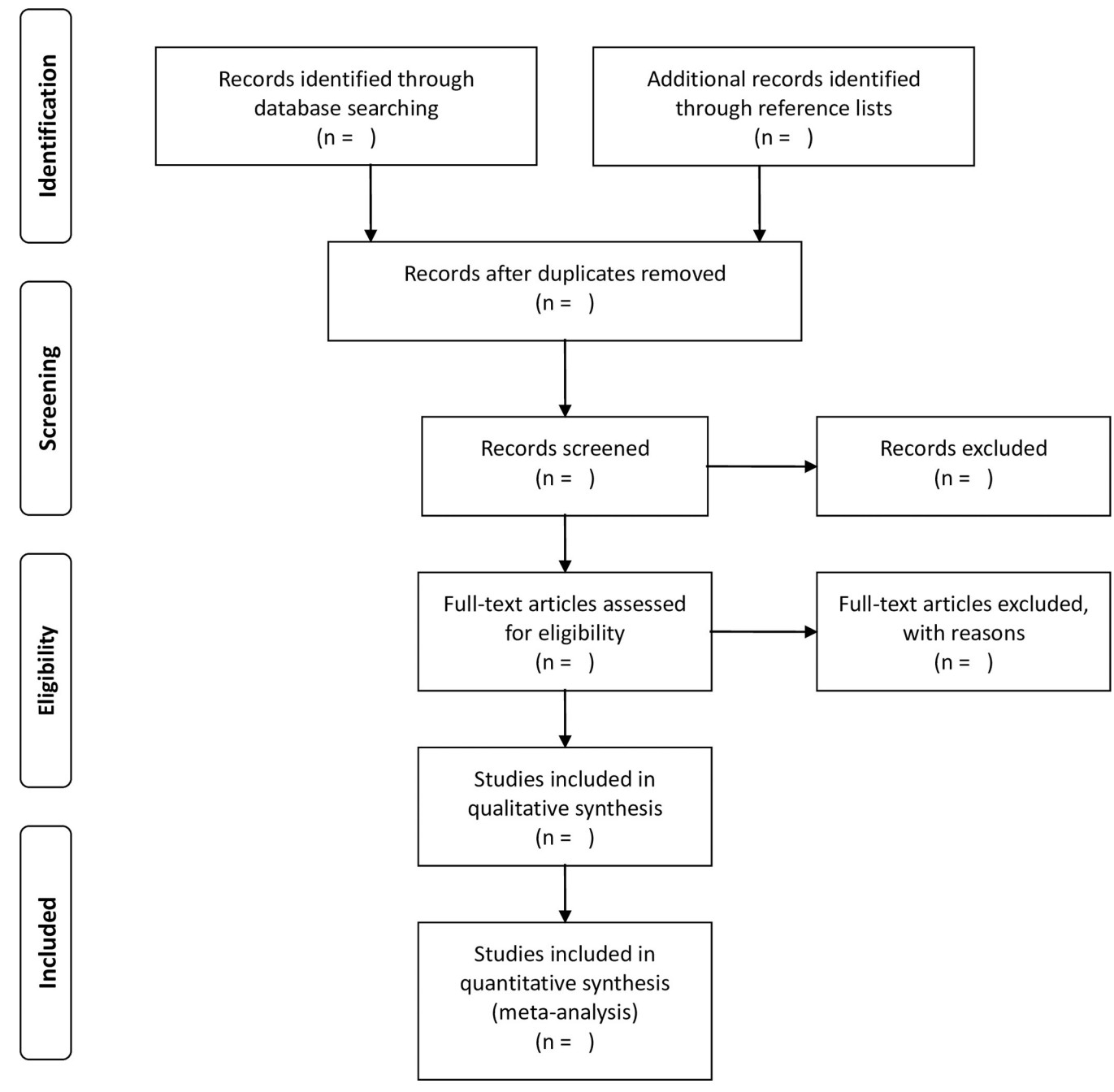

**Fig 1. PRISMA flow diagram.**

### Selection criteria

Inclusion criteria for this meta-analysis included: (1) included adult (>18 years) patients; (2) quantified epicardial fat volume using CT or MRI; (3) measured cardiac structure and function on echocardiography, CT or MRI or assessed cardiovascular outcomes; and (4) described the relation of epicardial fat volume with cardiac structure and function or cardiovascular outcomes.

## Outcomes

The primary outcome will be cardiac structure and function. The secondary outcome will be cardiovascular events including death from cardiovascular causes, hospitalization for HF, non-fatal myocardial infarction, and unstable angina. In the measures of cardiac structure, LV mass and left atrial volume will be extracted. In the measures of LV systolic function, LV ejection fraction and early systolic mitral annular velocity (s') will be extracted. In the measures of LV diastolic function, early diastolic mitral annular velocity (e') and the ratio of early diastolic mitral inflow to annular velocities (E/e') will be extracted given the linear relationship with LV diastolic dysfunction grade.

## Data extraction

Two reviewers (HF and TG) will independently extract relevant data from retrieved studies, including author, study design, study time, country, number of participants, baseline characteristics (age, sex, body mass index, epicardial fat volume, used modalities for assessing epicardial fat volume and cardiac structure and function, etc), clinical outcomes (cardiac structure and function and cardiovascular events), and information on the methodological quality (selection of cohorts, assessment of outcome, etc). Disagreements will be resolved by consensus. We will contact the corresponding author of eligible studies when insufficient information is available to perform our meta-analysis.

## Quality assessment

The quality of observational studies will be evaluated by Newcastle-Ottawa Scale tool (http://www.ohri.ca/programs/clinical_epidemiology/oxford.asp).

## Statistical analysis

To assess the association of pericardial fat with cardiac structure and function, correlation coefficients between pericardial fat volume and indices of cardiac structure and function will be synthesized. To assess the association of epicardial fat with cardiovascular outcomes, hazard ratios adjusted by variables of age and sex at least will be synthesized. For each outcome, heterogeneity will be assessed using the Cochran's Q and $I^2$ statistic; for the Cochran's Q and $I^2$ statistic, a p value of $<0.1$ and $I^2>50\%$, will be considered significant, respectively. When there is significant heterogeneity, the data will be pooled using a random-effects model, otherwise a fixed-effects model will be used. When there are more than 10 studies included, publication bias will be assessed graphically using a funnel plot and mathematically using Egger test. For these analyses, Comprehensive Meta Analysis Software version 2 (Biostat, Englewood, NJ, USA) and STATA 16 software (Stata Corp LP, TX, USA) will be used.

For outcomes with insufficient data, a systematic narrative synthesis will be provided with information presented in the text and tables to summarize and explain the characteristics and findings of the included studies. The narrative synthesis will explore the relationship and findings both within and between the included studies, in compliance with the guidance from the Centre for Reviews and Dissemination [11].

## Sensitivity analysis

Subgroup analysis stratified by study design (prospective or retrospective), modalities for quantifying epicardial fat volume (CT or MRI), and baseline patient characteristics (patients with cardiovascular diseases or general population, obese or not, elderly [$\geq$65 years] or not) will be performed. Meta-regression will be used to determine whether the association of

increased epicardial fat with outcomes will be confounded by baseline clinical characteristics (age and body mass index).

## Ethical issues

This meta-analysis is a literature study. Ethical approval is not required because this meta-analysis will not involve any subject directly.

## Discussion

There are plausible mechanisms to hypothesize that increased epicardial fat may be associated with cardiac remodeling and dysfunction as well as cardiovascular outcomes. First, visceral adipose tissue regulates the secretion of systemic adipokines; increased epicardial fat stimulates leptin secretion and downregulates adiponectin [1]. In animal and human studies, decreased levels of adiponectin and increased levels of leptin have been reported to be associated with LV hypertrophy and LV diastolic dysfunction [12–16]. Second, we and others previously reported the association of epicardial fat volume with aortic stiffness [5, 17, 18]. Aortic stiffening resulting from increased epicardial fat enhances pulse wave velocity and produces an earlier wave reflection in the central aorta, thereby increasing the pressure in late systole and decreasing the pressure in diastole [19]. With an increase in late-systolic load due to aortic stiffening, LV relaxation becomes delayed and incomplete, contributing to persistent pressure generation at end-diastole [20, 21]. Furthermore, decreased aortic pressure in diastole compromises coronary perfusion and may cause subendmyocardial ischemia [22], further enhancing LV diastolic dysfunction, especially in patients with coronary artery disease [23]. Finally, reduced circulating natriuretic peptides (NPs) levels due to increased expression of NP clearance receptors in adipose tissue may be a link between increased epicardial fat and abnormal cardiac structure and function as well as cardiovascular events. In addition to their natriuretic and vasorelaxant properties, NPs prevent cardiac hypertrophy and fibrosis, thereby counteracting the development and progression of HF [24]. The results of our systematic review and meta-analysis will provide evidence regarding the clinical usefulness of epicardial fat assessment.

## Supporting information

**S1 Checklist. PRISMA-P (Preferred Reporting Items for Systematic review and Meta-Analysis Protocols) 2015 checklist: Recommended items to address in a systematic review protocol.**
(DOC)

## Author Contributions

**Conceptualization:** Hidekatsu Fukuta, Takeshi Kamiya.

**Data curation:** Hidekatsu Fukuta, Toshihiko Goto.

**Methodology:** Hidekatsu Fukuta.

**Writing – original draft:** Hidekatsu Fukuta.

**Writing – review & editing:** Hidekatsu Fukuta, Toshihiko Goto, Takeshi Kamiya.

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
