## [Decision Letter · Decision Letter 0]

27 Jan 2023

PONE-D-22-25832Association of epicardial fat with cardiac structure and function and cardiovascular outcomes: a protocol for systematic review and meta-analysisPLOS ONE

Dear Dr. Fukuta,

Thank you for submitting your manuscript to PLOS ONE. After careful consideration, we feel that it has merit but does not fully meet PLOS ONE’s publication criteria as it currently stands. Therefore, we invite you to submit a revised version of the manuscript that addresses the points raised during the review process.

We look forward to receiving your revised manuscript.

Kind regards,

Roberto Magalhães Saraiva, MD, PhD

Academic Editor

PLOS ONE

and https://journals.plos.org/plosone/s/file?id=ba62/PLOSOne_formatting_sample_title_authors_affiliations.pdf.

Reviewers' comments:

Reviewer's Responses to Questions

**Comments to the Author**

1. Does the manuscript provide a valid rationale for the proposed study, with clearly identified and justified research questions?

Reviewer #1: Yes

Reviewer #2: Yes

2. Is the protocol technically sound and planned in a manner that will lead to a meaningful outcome and allow testing the stated hypotheses?

Reviewer #1: Yes

Reviewer #2: No

3. Is the methodology feasible and described in sufficient detail to allow the work to be replicable?

Reviewer #1: Yes

Reviewer #2: No

4. Have the authors described where all data underlying the findings will be made available when the study is complete?

Reviewer #1: Yes

Reviewer #2: No

5. Is the manuscript presented in an intelligible fashion and written in standard English?

Reviewer #1: Yes

Reviewer #2: Yes

6. Review Comments to the Author

You may also provide optional suggestions and comments to authors that they might find helpful in planning their study.

Reviewer #1: The authors provide an interesting and potential important manuscript describing "Association of epicardial fat with cardiac structure and function and cardiovascular

outcomes: a protocol for systematic review and meta-analysis", The main issues concerning this paper are those concerning the potential associations between epicardial fat and cardiac structure and function and cardiovascular outcomes

There are some weak points that need to be addressed by the authors.

Minor

1.In general, excess adipocytes have some negative effects on cardiovascular disease, but adipocytes, as an endocrine cell, can also secrete molecules such as adiponectin, omentin, and others that are beneficial for cardiovascular conditions. Adipose secreted factors in the context of non obesity, may play a protective role against cardiovascular. Patients older than 18 years were selected in the text for the determination of epicardial fat volume using MRI or CT and were not qualitative about whether the selected subjects were obese or not. So it is recommended to add grouping in this respect.

2.On the other hand, the age factor has a direct effect on mortality and prognosis, in order to better represent the effect of epicardial fat on CVD, the age grouping can be further increased under the premise that the data collected in the articles are comprehensive.

Reviewer #2: To my knowledge, there is no systematic literature review and meta-analysis that evaluates the association between epicardial fat and cardiac remodeling/cardiovascular events.

However, several criticisms should to be addressed:

- In the abstract, it is not clear the main aim of the study. In the secondary outcome, it is incorrect to consider the cardiac structure and dysfunction within the cardiovascular events. Furthermore, major outcomes, such as cardiovascular death and hospitalizations for HF, are already considered in the primary outcome of the study. Cardiac structure and function seems to be the main focus of the work, reading the title, but the primary objective concerns instead cardiovascular events. Clarify this aspect better.

- More details are required in the method section. Who and how many researchers will the literature search be performed? What is the combination of keywords used in the literature search? Will a hand search of references be performed upon completion? How often will the literature search be repeated?

- The authors plan to include only studies with measured cardiac structure and function on echocardiography. Could it also be useful to include studies that have measured cardiac structure and function with other less common methods, in order to have a more complete view of the literature?

- Regarding cardiovascular outcome, it is not clear why the authors want to focus their work only on heart failure?

- More details regarding the data extraction would be required.

- In the discussion section, among the possible mechanisms linking visceral fat and cardiac dysfunction, the authors cannot overlook the role of cardiac natriuretic peptides (see PMID: 36430893).

7. PLOS authors have the option to publish the peer review history of their article (what does this mean?). If published, this will include your full peer review and any attached files.

Reviewer #1: No

Reviewer #2: No

---

## [Author Response · Author response to Decision Letter 0]

13 Feb 2023

Response to Reviewers

Reviewer #1: The authors provide an interesting and potential important manuscript describing "Association of epicardial fat with cardiac structure and function and cardiovascular

outcomes: a protocol for systematic review and meta-analysis", The main issues concerning this paper are those concerning the potential associations between epicardial fat and cardiac structure and function and cardiovascular outcomes

There are some weak points that need to be addressed by the authors.

We are grateful for the comments of the reviewer, which have helped us improve our manuscript.

1.In general, excess adipocytes have some negative effects on cardiovascular disease, but adipocytes, as an endocrine cell, can also secrete molecules such as adiponectin, omentin, and others that are beneficial for cardiovascular conditions. Adipose secreted factors in the context of non obesity, may play a protective role against cardiovascular. Patients older than 18 years were selected in the text for the determination of epicardial fat volume using MRI or CT and were not qualitative about whether the selected subjects were obese or not. So it is recommended to add grouping in this respect.

As indicated, we have added the description that the subgroup analysis stratified by obese or not will be performed (page 10, para. 3, lines 1-4) and that meta-regression will be used to determine whether the association of increased epicardial fat with outcomes will be confounded by body mass index (page 10, para. 3, lines 4-6).

2.On the other hand, the age factor has a direct effect on mortality and prognosis, in order to better represent the effect of epicardial fat on CVD, the age grouping can be further increased under the premise that the data collected in the articles are comprehensive.

As indicated, we have added the description that the subgroup analysis stratified by elderly or not will be performed (page 10, para. 3, lines 1-4) and that meta-regression will be used to determine whether the association of increased epicardial fat with outcomes will be confounded by age (page 10, para.3, lines 4-6).

 

Reviewer #2: To my knowledge, there is no systematic literature review and meta-analysis that evaluates the association between epicardial fat and cardiac remodeling/cardiovascular events.

However, several criticisms should to be addressed:

We are grateful for the excellent review.

- In the abstract, it is not clear the main aim of the study. In the secondary outcome, it is incorrect to consider the cardiac structure and dysfunction within the cardiovascular events. Furthermore, major outcomes, such as cardiovascular death and hospitalizations for HF, are already considered in the primary outcome of the study. Cardiac structure and function seems to be the main focus of the work, reading the title, but the primary objective concerns instead cardiovascular events. Clarify this aspect better.

As indicated, we have revised the definitions of the primary and secondary outcomes (page 8, para.1, lines 1-3).

- More details are required in the method section. Who and how many researchers will the literature search be performed? What is the combination of keywords used in the literature search? Will a hand search of references be performed upon completion? How often will the literature search be repeated?

As indicated, we have revised the description of search strategy (page 6, para.2, carrying over to next page) and have added flow diagram (Fig. 1).

- The authors plan to include only studies with measured cardiac structure and function on echocardiography. Could it also be useful to include studies that have measured cardiac structure and function with other less common methods, in order to have a more complete view of the literature?

As suggested, we have revised the description of selection criteria (page 7, para. 3, lines 2-4).

- Regarding cardiovascular outcome, it is not clear why the authors want to focus their work only on heart failure?

As indicated, we have revised the definition of cardiovascular outcomes (page 8, para. 1, lines 1-3).

- More details regarding the data extraction would be required.

As indicated, we have revised the description of data extraction (page 8, para. 2, carrying over to next page). 

- In the discussion section, among the possible mechanisms linking visceral fat and cardiac dysfunction, the authors cannot overlook the role of cardiac natriuretic peptides (see PMID: 36430893).

As indicated, we have discussed the role of natriuretic peptides for the possible mechanisms linking visceral fat and cardiac dysfunction (page 12, para.1, lines 4-9).

---

## [Decision Letter · Decision Letter 1]

9 Mar 2023

Association of epicardial fat with cardiac structure and function and cardiovascular outcomes: a protocol for systematic review and meta-analysis

PONE-D-22-25832R1

Dear Dr. Fukuta,

We’re pleased to inform you that your manuscript has been judged scientifically suitable for publication and will be formally accepted for publication once it meets all outstanding technical requirements.

Kind regards,

Roberto Magalhães Saraiva, MD, PhD

Academic Editor

PLOS ONE

Additional Editor Comments (optional):

Reviewers' comments:

Reviewer's Responses to Questions

**Comments to the Author**

1. Does the manuscript provide a valid rationale for the proposed study, with clearly identified and justified research questions?

Reviewer #1: Yes

Reviewer #2: Yes

2. Is the protocol technically sound and planned in a manner that will lead to a meaningful outcome and allow testing the stated hypotheses?

Reviewer #1: Yes

Reviewer #2: Yes

3. Is the methodology feasible and described in sufficient detail to allow the work to be replicable?

Reviewer #1: Yes

Reviewer #2: Yes

4. Have the authors described where all data underlying the findings will be made available when the study is complete?

Reviewer #1: Yes

Reviewer #2: Yes

5. Is the manuscript presented in an intelligible fashion and written in standard English?

Reviewer #1: Yes

Reviewer #2: Yes

6. Review Comments to the Author

You may also provide optional suggestions and comments to authors that they might find helpful in planning their study.

Reviewer #1: All the problems have been solved. The article provides the association of empirical fat with cardiac structure and function and cardiovascular outcomes, which is important for the new research

Reviewer #2: The authors satisfied my previous requests and now the manuscript is clearer. I have no further suggestions.

7. PLOS authors have the option to publish the peer review history of their article (what does this mean?). If published, this will include your full peer review and any attached files.

Reviewer #1: No

Reviewer #2: No

---

## [Editor Report · Acceptance letter]

5 Apr 2023

PONE-D-22-25832R1 

Association of epicardial fat with cardiac structure and function and cardiovascular outcomes: a protocol for systematic review and meta-analysis 

Dear Dr. Fukuta:

I'm pleased to inform you that your manuscript has been deemed suitable for publication in PLOS ONE. Congratulations! Your manuscript is now with our production department. 

Kind regards, 

on behalf of

Dr. Roberto Magalhães Saraiva 

Academic Editor

PLOS ONE